# Accuracy of Intra-Axial Brain Tumor Characterization in the Emergency MRI Reports: A Retrospective Human Performance Benchmarking Pilot Study

**DOI:** 10.3390/diagnostics14161791

**Published:** 2024-08-16

**Authors:** Aapo Sirén, Elina Turkia, Mikko Nyman, Jussi Hirvonen

**Affiliations:** 1Department of Radiology, Turku University Hospital, and University of Turku, Kiinamyllynkatu 4-8, 20520 Turku, Finland; 2Medical Imaging Center, Department of Radiology, Tampere University Hospital, and Tampere University, 33520 Tampere, Finland

**Keywords:** emergency radiology, neuroradiology, brain tumor, magnetic resonance imaging, diagnostic accuracy, glioma, brain metastasis, brain lymphoma

## Abstract

Demand for emergency neuroimaging is increasing. Even magnetic resonance imaging (MRI) is often performed outside office hours, sometimes revealing more uncommon entities like brain tumors. The scientific literature studying artificial intelligence (AI) methods for classifying brain tumors on imaging is growing, but knowledge about the radiologist’s performance on this task is surprisingly scarce. Our study aimed to tentatively fill this knowledge gap. We hypothesized that the radiologist could classify intra-axial brain tumors at the emergency department with clinically acceptable accuracy. We retrospectively examined emergency brain MRI reports from 2013 to 2021, the inclusion criteria being (1) emergency brain MRI, (2) no previously known intra-axial brain tumor, and (3) suspicion of an intra-axial brain tumor on emergency MRI report. The tumor type suggestion and the final clinical diagnosis were pooled into groups: (1) glial tumors, (2) metastasis, (3) lymphoma, and (4) other tumors. The final study sample included 150 patients, of which 108 had histopathological tumor type confirmation. Among the patients with histopathological tumor type confirmation, the accuracy of the MRI reports in classifying the tumor type was 0.86 for gliomas against other tumor types, 0.89 for metastases, and 0.99 for lymphomas. We found the result encouraging, given the prolific need for emergency imaging.

## 1. Introduction

Imaging in emergency departments has increased tremendously over the past few decades [1,2,3]. This includes head imaging [4], with not only computed tomography (CT), but also a growing number of requests for advanced studies, e.g., magnetic resonance imaging (MRI) [5]. A previously unknown brain tumor is an uncommon but not negligible finding in emergent head imaging [6], and any radiologist reporting emergency head imaging might discover a brain tumor. The most important task for the radiologist reporting the study is to point out complications requiring urgent treatment, e.g., hydrocephalus or intracranial hemorrhage. Primary central nervous system (CNS) tumors (glial or non-glial brain tumors), metastasis, and lymphoma require different approaches to the diagnostic workup, and the primary assumption of the tumor type guides the next steps [7,8,9]. The exact tumor characterization cannot be made based only on imaging. Still, imaging, especially MRI, is an integral part of the whole.

Primary CNS tumors are classified according to the WHO Classification of Tumors of the Central Nervous System, the latest update being the fifth edition, published in 2021 [10]. The occurrence of different brain tumor types differs by many factors, especially by age. Among children and adolescents, the most common primary brain tumor is a pilocytic astrocytoma, while the most common aggressive brain tumor is medulloblastoma. Compared to adults, children bear a higher risk for embryonal tumors and germ cell tumors [11,12]. In the adult population, the most common CNS tumor is meningioma, an extra-axial tumor. Of the intra-axial primary brain tumors in adults, the most common is glioblastoma, an aggressive glial tumor WHO grade 4 tumor. Still, the most common intra-axial brain tumor in adults is metastasis [13]. Brain metastases have been found to be as much as ten-fold more common than primary brain tumors [13].

Brain lymphoma is divided into secondary CNS lymphoma as an involvement of systemic lymphoma and primary central nervous system lymphoma [14]. Both are uncommon, but primary CNS lymphoma is even more uncommon, with the estimated incidence for primary central nervous system lymphoma in Finland being 0.68/100,000 person-years [15]. Lymphomas may affect intra- and extra-axial structures of the CNS, but the most common imaging presentation in the primary CNS lymphoma is a mass lesion in the brain parenchyma [14]. This study focuses on intra-axial CNS tumors because extra-axial tumors often have a distinct appearance and different workup strategies compared to intra-axial ones.

Some intra-axial tumors, e.g., pilocytic astrocytoma, usually have characteristic imaging features [16]. However, the most common intra-axial tumors in the adult population—metastasis and glioblastoma—may demonstrate very similar imaging features. They often present as ring-enhancing tumors with central necrosis and peritumoral edema. The presence of multiple tumors is often considered to favor metastatic etiology, and a solitary tumor is more often glioblastoma than metastasis. Still, solitary cerebral metastases and multifocal and multicentric glioblastomas exist. Even the differential between a tumor and tumor-like lesions, including infection or inflammation, can sometimes be challenging [17]. Imaging features of conventional and advanced MRI have been extensively studied to improve diagnostic accuracy in differentiating these pathologies [16,17,18,19,20]. In recent years, the scientific literature about artificial intelligence (AI) and radiomics in brain tumor characterization has rapidly increased [21,22,23,24]. Still, the knowledge of the real-life diagnostic accuracy of brain tumor characterization in MRI is surprisingly scarce. That is, currently, there is no properly defined benchmarking for the diagnostic accuracy of human performance in the task for which AI applications are already being developed.

Our emergency radiology department has had an MRI machine dedicated to emergency imaging since 2013. The brain is the most frequent subject of studies performed with emergency MRI. Hence, new brain tumors are occasionally seen. Often, a head CT has already raised the possibility of a tumor, but sometimes MRI is the first-line imaging modality. With this study, we aimed to assess the accuracy of the radiological magnetic resonance imaging reports in the emergency department regarding the intra-axial tumor type. Our hypothesis and practical rationale were that the radiologist is able to characterize the tumor with clinically acceptable accuracy in the emergency department to allow for appropriate follow-up strategies while avoiding unnecessary studies.

## 2. Materials and Methods

We retrospectively reviewed the charts of patients who underwent a brain MRI examination at our Emergency Radiology Department between 1 April 2013 and 31 January 2019. Our hospital is a tertiary care referral center for approximately 470,000 people. The inclusion criteria for the study sample were (1) emergency brain magnetic resonance imaging, (2) no previously diagnosed intra-axial brain tumor, and (3) suspicion of an intra-axial tumor on a primary emergency MRI report. The patients with (1) no final clinical diagnosis in the patient charts and (2) no specified suggestion of a tumor type on a primary MRI report were excluded.

The radiology information system (RIS) was reviewed to extract imaging reports with MRI and CT findings and the training level of a reporting radiologist. The MRI referrals were sought for information on potential previously known malignant diseases. Medical records were reviewed for demographic variables, symptoms, pathology reports, and the final clinical diagnosis. Only histopathological analysis of the brain tumor was taken into account, although in many cases, the final clinical diagnosis was partly based on the histological sample achieved from the suspected primary tumor or from a metastasis outside the brain. Regarding both radiology reports and final clinical diagnosis, the tumors were pooled into four groups: (1) glial tumors, (2) metastasis, (3) lymphoma, and (4) other intra-axial tumors, including non-glial primary central nervous system tumors.

Either a neurologist on call or another emergency physician, after consulting the neurologist, referred the MRI scans. Magnetic resonance imaging was performed in the emergency radiology department using a Philips Ingenia 3-tesla machine with a Philips dStream head coil system (Philips Healthcare, Best, The Netherlands). The routine brain tumor MRI protocol included the following sequences: axial T2-weighted, axial T2 fluid attenuation inversion recovery (FLAIR), isotropic 3D FLAIR, axial diffusion-weighted imaging (DWI), axial susceptibility-weighted (SWI) sequences, and isotropic 3D turbo spin echo T1-weighted without and with gadolinium enhancement (Appendix A). No specific preprocessing was performed before the image review. Magnetic resonance spectroscopy (MRS) and dynamic contrast-enhancement perfusion MRI (PWI) were performed in 71 (47%) and 67 (45%) of the cases, although they were noted on an MRI report only in 43 (29%) and 46 (31%) of the cases, respectively. 

A retrospective image review was not performed because we were interested in the emergency radiologists’ performance in a real-life setting. The experience levels of the radiologist giving the primary MRI report were as follows: a radiologist in training (with at least three years of experience in radiology) reported 2/150 (1%) studies, a board-certified radiologist (at least five years of experience in radiology) reported 43/150 (29%) studies, and a fellowship-trained neuroradiologist (at least seven years of experience in radiology) reported 105/150 (70%) studies. Most studies (97/150, 65%) were reported between 7 am and 4 pm, but one-third of the reports were given outside the office hours between 4 pm and 10 pm (53/150, 35%). None of the MRI reports included in the study were given during the night, between 10 pm and 7 am, apparently because during the night MRI is available only for the most critical cases for which CT is not enough to assess the need for immediate treatment.

The results are expressed as the number of cases (n), percentage, mean, median, and range. For sensitivity, specificity, positive predictive value, negative predictive value, accuracy, and the F1-score, the 95% confidence intervals were calculated. Proportions of categorical variables were compared with the Pearson Chi-square (*X*^2^) test. *p*-values  <  0.05 were considered statistically significant. The statistical analyses were performed using the IBM SPSS Statistics Package for Mac (version 29, IBM Corporation, Armonk, NY, USA).

We obtained permission from the hospital district board, but institutional ethical review board approval and written patient consent were not needed for this retrospective study.

## 3. Results

We found 168 patients meeting the inclusion criteria. After excluding six patients without any final clinical diagnosis in the patient charts and twelve patients with only descriptive primary MRI reports without a tumor type suggestion, the final study sample included 150 patients. The basic demographics of the study population are presented in Table 1. The age range was wide, from 4 to 89 years, but most patients were in late adulthood, their mean age being 62 years and median age being 65 years. The sex ratio was almost even. The symptoms and clinical findings described in the medical reports as an indication for the MRI are presented in Figure 1. There was a very high variance in the reported duration of the symptoms. In 34 cases, the symptoms had a sudden onset, leading to immediate evaluation at the emergency department, but the longest reported time from the first appearance of the symptoms leading to the brain MRI was two years, the median being ten days. Two tumors were found incidentally. The first of these incidental tumors was found on a head CT performed because of symptomatic head trauma, and the second on an MRI performed because of a symptomatic ischemic stroke.

In 108/150 patients (72%), histopathological confirmation of the brain tumor was found in the medical records (Table 1). Overall, the primary suggestion of a tumor type in an emergency MRI report corresponded with the final clinical diagnosis in 129/150 (86%) cases (Table 2). In the cases with histopathological confirmation, the primary suggestion of the tumor type corresponded with the final diagnosis in 91/108 (83%) of the cases, with no statistically significant differences in the proportions of correct tumor type suggestions between the groups with and without histopathological confirmation (*p* = 0.190, *X*^2^ = 2278, df = 1). No statistically significant differences between the proportions of the correct primary diagnostic suggestions were found when comparing the cases with known malignancy to those without previously known malignancy. We found no statistically significant differences when comparing reports of general radiologists and neuroradiologists or between reports within and outside office hours. Neither the inclusion of PWI nor MRS in the primary report had a statistically significant effect on the accuracy of the primary suggestion regarding the tumor type. Lymphomas seemed easiest to categorize accurately, and 10/11 (91%) cases with lymphoma as a final clinical diagnosis were correctly suggested in the emergency MRI report.

Sensitivity, specificity, positive predictive value (PPV), negative predictive value (NPV), accuracy, and F1-score of the MRI reports were calculated for the histopathologically confirmed gliomas, metastases, and lymphomas. The results are presented in Table 3. In general, on-call emergency radiologists were highly accurate in predicting the primary histological tumor type, lymphoma having the highest levels of discrimination from other primary types.

In 29/150 (19%) of the patients, more than one tumoral lesion was seen. Of these cases, 24/27 (89%) were primarily suggested to be metastases, 3/27 (11%) were suggested to be multifocal or multicentric gliomas, 1/27 (4%) were suggested to be multiple hemangioblastomas, and 1/27 (4%) were suggested to be multiple cerebral parenchymal lesions of lymphoma. Of the MRI reports considering patients with multiple tumors, 26/29 (90%) had a primary tumor type suggestion concordant with the final clinical diagnosis. All four cases with incorrect tumor type suggestions on emergency MRI reports were histopathologically confirmed multifocal or multicentric gliomas that were primarily suspected of being metastases.

Computed tomography preceding MRI was used in 127/150 cases (85%), and in 122/127 (96%) cases, a tumor was suspected in the CT report. Of the cases with no suspicion of tumor in the CT report, two (1.6%) were cases with a tumor hemorrhage hiding the underlying tumor, two cases (1.6%) were misinterpreted as an ischemic lesion, and one case (0.8%) with a supratentorial intra-axial metastasis was simply missed.

Table 4 presents the basic demographics of the patients with final clinical diagnoses of glioma, metastasis, and lymphoma. The patients with gliomas were younger than patients with metastasis, lymphoma patients being the oldest. History of malignancy was much more common among the patients whose final clinical diagnosis was brain metastasis compared to patients with a diagnosis of glioma or lymphoma. No statistically significant differences in the proportions of sexes in groups with different tumors were observed.

## 4. Discussion

In this sample of 150 patients with previously unknown brain tumors assessed with emergency MRI, the diagnostic accuracy was on a level we consider clinically acceptable. Considering the high and increasing workload in the emergency radiology departments, the results are encouraging.

Of all cases in the study sample, 129/150 (86%) of the primary tumor type suggestions in the radiology report corresponded with the final clinical diagnosis. With the cases having histopathological correction of the tumor type, the respective numbers were 91/108 (83%). Even though the primary diagnostic suggestion would have guided the final clinical diagnosis if the histopathological analysis had not been performed, the differences between the correct tumor type suggestion in the groups with and without histopathological analysis were not statistically significant. The sensitivity of the MRI reports regarding broad tumor type characterization was 0.86 (95% confidence interval 0.76–0.93) for the gliomas, 0.88 (95% CI 0.61–0.98) for metastases, and 0.89 (95% CI 0.52–1.00) for the lymphomas, with specificity being 0.87 (95% CI 0.70–0.96), 0.89 (95% CI 0.81–0.95), and 1.00 (95% CI 0.96–1.00), respectively (Table 3). 

Lymphomas were identified with very high accuracy, although the total amount of histopathologically confirmed lymphomas in our sample was small, with only nine cases. Lymphomas have a characteristic feature of high cellularity, demonstrating high signal in diffusion-weighted imaging (DWI) and low values in apparent diffusion coefficient (ADC) maps [14]. This is an uncommon finding in other intra-axial tumors, often making lymphoma detection relatively straightforward, combined with characteristic shape, contrast enhancement patterns, and location of lymphomas. Cerebral abscesses express similar DWI characteristics, but other imaging features and clinical findings are usually distinct. Of the cases with multiple brain tumors, 26/29 (90%) had the primary tumor type suggestion on emergency magnetic imaging reports corresponding with the final clinical diagnosis. All misinterpreted cases with multiple tumors were suggested as being metastases but turned out to be histopathologically confirmed to be multifocal or multicentric gliomas. In all cases with multiple tumors in which the primary emergency MRI report suggested other pathology than metastases (multifocal or multicentric glioma, multiple hemangioblastomas, multiple lymphoma lesions of brain parenchyma), the primary tumor type suggestion concorded with the final clinical diagnosis. Differentiation of multifocal or multicentric glioblastoma and metastases is not always straightforward and sometimes not possible at all [25].

Correct tentative tumor type characterization reduces unnecessary steps in the definitive diagnostic pathway. A suspicion of metastasis or lymphoma should lead to a search for the primary tumor and determination of disease extent, while in gliomas, further imaging is not routinely needed. Figure 2 demonstrates two examples of glioblastomas, of which another one was initially suggested to represent a solitary metastasis. This led to unnecessary thoracoabdominal computed tomography to search for a primary tumor. Unnecessary scans mean increased expenses, radiation exposure, and delayed treatment. Regardless, knowing the difficulties in discriminating glioma from metastasis based on MRI only, the threshold for performing thoracoabdominal CT in case of newly detected brain tumors must remain low.

Incorrect tumor-type suggestions may also lead to inappropriate treatment. Unfortunately, symptoms, clinical findings, and imaging characteristics may be misleading sometimes. In our study sample, one patient was considered to have a glioma, and with a multidisciplinary team consensus, the tumor was surgically resected without a preceding brain biopsy. However, the histopathological examination revealed the lesion to represent an uncommon demyelinating disease, Baló concentric sclerosis, which does not require or benefit from operative treatment. One meningioma with a very atypical MRI appearance was interpreted to be a glioma and was surgically resected after a multidisciplinary team consensus. Even though the preoperative diagnosis was incorrect, the treatment turned out to be proper and curative. All the other 148/150 (99%) lesions interpreted to represent intra-axial neoplasms were intra-axial neoplasms. The patient with Baló concentric sclerosis was the only patient with a non-neoplastic tumor mimic, but mimics must be considered when assessing a tumoral lesion. Many infectious and inflammatory conditions, such as neuro toxoplasmosis, human immunodeficiency virus (HIV)-related encephalopathy or encephalitis, and progressive multifocal leukoencephalopathy (PML), can demonstrate MRI appearance similar to metastasis, lymphomas, or gliomas.

The MRI referrals in our study included some very common symptoms that very often lead to head imaging in the emergency department (Figure 1). The first imaging modality in the diagnostic workup of a neurological patient is most often a head CT. In our study, the radiologists suspected a tumor in 96% (122/127) of the cases with a head CT preceding emergency MRI. In four misinterpreted CT scans, the lesion was primarily suggested to be a primary hemorrhage or ischemic stroke, but the discrepancy between the clinical scenario and computed tomography report led to an MRI to further characterize the lesion. One tumor was simply missed on the primary CT report. These kinds of mistakes are unfortunate and should not happen, yet they inevitably occur sometimes over a long period of time in a busy unit. The risk for radiological errors is especially significant at night and under a high workload [26]. A hemorrhagic or ischemic stroke with unusual clinical presentation or atypical imaging findings is, practically without exception, further reviewed with a follow-up computed tomography or subsequent magnetic resonance imaging, ultimately leading to a tumor diagnosis [27,28].

Brain tumor characterization is usually infeasible based on computed tomography alone and can be even challenging with magnetic resonance imaging. Sometimes, even tumoral and non-tumoral pathologies are difficult to distinguish from each other. Recently, novel MRI techniques [19,20], artificial intelligence (AI), and radiomics [21,22,23,24] have been intensively studied to enhance brain tumor characterization. However, applications that are feasible in clinical practice are still lacking. There are various medico-legal issues to be solved before artificial intelligence applications are expected to be used in daily radiological practice [29]. Also, in our sample, the accuracy of emergency MRI reports based mostly on conventional brain MRI sequences was comparable to that of contemporary AI models. A meta-analysis by pooling studies that assessed AI models trained to differentiate glioma from metastasis reported both pooled sensitivity and specificity to be 0.84 [30], whereas a study in which an AI model was trained with a large dataset to distinguish several types of brain tumors reported overall sensitivity of 0.73–0.89 and specificity of 0.85–0.97, depending on the testing data [22]. Given that AI approaches are increasingly being studied and utilized for this task, accurate benchmarking of human performance, as done in the current work, is critical to be able to fully evaluate the added value of these automated methods.

Our study has certain limitations. The most obvious are the inherent biases of the retrospective study. There was heterogeneity in the thoroughness of radiology reports and the clinical data. All patients did not undergo a brain tumor biopsy or brain tumor resection. The first putative radiological diagnosis has probably impacted the final clinical diagnosis among the cases without histopathological tumor sample analysis, even though most patients were scanned subsequently during follow-up, and the final clinical diagnosis was based on the comprehensive judgment, often with a multidisciplinary team. This point is especially relevant for metastatic disease, where brain tissue sampling may not have been scheduled. The sample size in our study was relatively small, and other tumors than glioma and metastasis were infrequent. In the future, larger-scale multicenter studies would be useful. The radiologists working in our emergency radiology department might have more experience with brain MRIs because we have had dedicated emergency MRIs for a long time. This might also explain why there was no statistically significant difference between the reports by fellowship-trained neuroradiologists and other radiologists. However, we see that our results represent the reality of a small tertiary care hospital with a well-organized emergency radiology department.

## 5. Conclusions

In conclusion, our study suggests that radiologists are able to putatively characterize intra-axial brain tumors with MRI in the emergency department with good accuracy. Whether we like it or not, the requests for emergent imaging seem to keep on increasing, leading to a broader spectrum of findings in emergency imaging. The radiologist’s role is to guide the diagnostic workup to the right tracks, though the definitive brain tumor classification is not to be made during the on-call shifts.

## Figures and Tables

**Figure 1 diagnostics-14-01791-f001:**
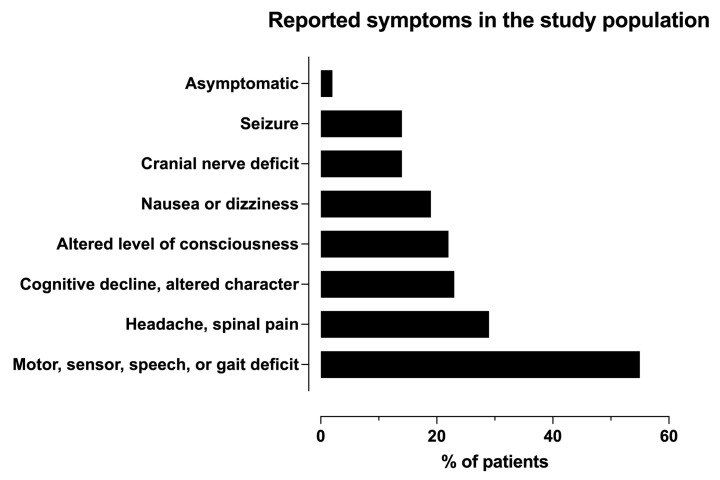
Symptoms described in the emergency MRI referral. All the symptoms mentioned in the MRI referrals as an indication for brain MRI were extracted and pooled. The symptoms in the study population were non-specific, similar to those of most neurological patients in the emergency departments. The most common were problems with motor function and speech.

**Figure 2 diagnostics-14-01791-f002:**
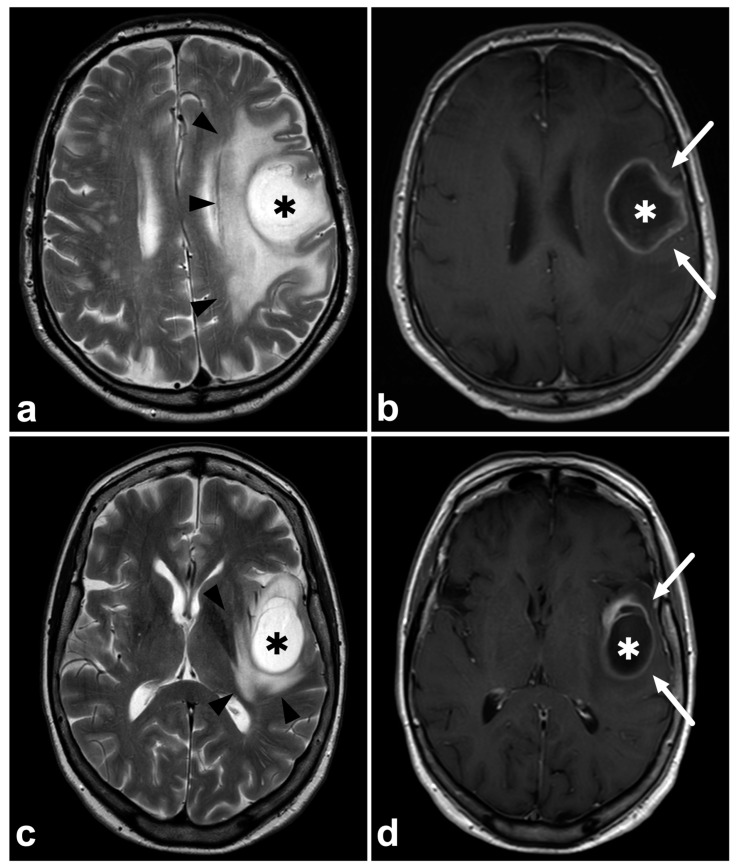
(**a**) Patient 1, axial T2-weighted MRI, (**b**) Patient 1, axial gadolinium-enhanced T1-weighted spin-echo MRI, (**c**) Patient 2, axial T2-weighted MRI, (**d**) Patient 2, axial gadolinium-enhanced T1-weighted spin-echo MRI. Two patients with a histopathologically confirmed glioblastoma. Patient 1 (upper row) was initially suggested to have a solitary cerebral metastasis (asterisks), primarily because of the rapid appearance of the tumor (the previous MRI six months earlier was unremarkable) and because of the extensive edema surrounding the tumor (black arrowheads). Patient 1 underwent a thoracoabdominal CT scan to detect the primary tumor and other metastases. Patient 2 (lower row) was initially suggested of having a glioblastoma (asterisks) and the patient was immediately directed to the neurosurgeon without unnecessary steps. The diagnosis of glioblastoma was later confirmed histopathologically. Both tumors demonstrate a homogenous, probably necrotic center (asterisks) surrounded by a gadolinium-enhancing rim (white arrows). The enhancing rim is surrounded by a T2-hyperintense zone consisting of poorly delineated tumor infiltration and vasogenic edema, sometimes inseparable from each other with conventional clinical imaging. Brain metastasis often represents very similar MRI findings compared to glioblastomas.

**Table 1 diagnostics-14-01791-t001:** Demographic characteristics, proposed tumor types in MRI reports, and final clinical diagnoses.

**Number of patients**	150
**Age, mean (SD), range, median**	62, (16.1), 4–89, 65
**Female, *n* (%)**	72 (48)
**History of malignancy**	***n* (%)**
Yes	37 (25)
No	113 (75)
**A primary suggestion of brain tumor type on MRI report**	***n* (%)**
Glioma ^1^	71 (47)
Metastasis	61 (41)
Lymphoma	11 (7)
Non-glial primary CNS tumor ^2^	7 (5)
**Final clinical diagnosis**	***n* (%)**
Glioma ^3^	81 (54)
Metastasis	52 (35)
Lymphoma	11 (7)
Non-glial primary CNS tumor ^4^	4 (3)
Non-tumoral lesion (Baló concentric sclerosis)	1 (0.6)
**Histopathological confirmation of a brain tumor**	***n* (%)**
Yes	108 (72)
No	42 (28)

^1^ Including one proposed subependymoma; ^2^ hemangioblastoma, medulloblastoma, pineocytoma; ^3^ Including two ependymomas ^4^ hemangioblastoma, medulloblastoma.

**Table 2 diagnostics-14-01791-t002:** Correct and incorrect suggestions of tumor types grouped by patient history, training of the reporting radiologist, and the time of the day the report was given.

	All Tumors			Tumors with Histopathological Confirmation		
	Correct Tumor Type Suggestion	Incorrect Tumor Type Suggestion	*p*-Value	Correct Tumor Type Suggestion	Incorrect Tumor Type Suggestion	*p*-Value
Known malignancy	33	4	0.598 ^1^	14	3	0.906 ^1^
No known malignancy	96	17		76	15	
General radiologist	34	9	0.193 ^1^	13	5	0.084 ^1^
Neuroradiologist	93	12		77	13	
Reported within office hours	82	15	0.624 ^1^	59	14	0.413 ^1^
Reported outside office hours	47	6		31	4	

^1^ *X*^2^ test.

**Table 3 diagnostics-14-01791-t003:** Accuracy of the tumor type characterization in the emergency MRI report using the histopathological diagnosis as the reference standard.

	Glioma vs. Other Pathologies	Metastasis vs. Other Pathologies	Lymphoma vs. Other Pathologies
Sensitivity (95% confidence interval)	0.86 (0.76–0.93)	0.88 (0.61–0.98)	0.89 (0.52–1.00)
Specificity (95% confidence interval)	0.87 (0.70–0.96)	0.89 (0.81–0.95)	1.00 (0.96–1.00)
PPV (95% confidence interval)	0.94 (0.87–0.98)	0.58 (0.43–0.72)	1.00 (0.63–1.00)
NPV (95% confidence interval)	0.71 (0.58–0.81)	0.98 (0.92–0.99)	0.99 (0.94–1.00)
Accuracy (95% confidence interval)	0.86 (0.78–0.92)	0.89 (0.81–0.94)	0.99 (0.95–1.00)
F1-score (95% confidence interval)	0.90 (0.81–0.99)	0.70 (0.46–0.94)	0.94 (0.74–1.0)

**Table 4 diagnostics-14-01791-t004:** Demographic characteristics in groups with different final clinical diagnoses.

	Glioma (*n* = 81)	Metastasis (*n* = 52)	Lymphoma (*n* = 11)	*p*-Value
Age, mean (SD), range	60 (18), 4–93	67 (11), 41–88	69 (7), 59–83	0.020 *
Female, *n* (%)	35 (43%)	31 (60%)	5 (46%)	0.177 **
Previously known malignancy, *n* (%)	4 (5%)	29 (56%)	3 (27%)	<0.001 **

* One-way ANOVA; ** *X*^2^ test.

## Data Availability

Data cannot be publicly shared because of national legislature on the privacy of patient data.

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
