# Peer review of "Accuracy of Intra-Axial Brain Tumor Characterization in the Emergency MRI Reports: A Retrospective Human Performance Benchmarking Pilot Study"

_diagnostics, 2024, doi:10.3390/diagnostics14161791_

Round 1

Reviewer 1 Report

Comments and Suggestions for Authors

High sensitivity and specificity of the MRI method, especially when using a 3 Tesla CT scanner, together with good experience of the radiologist certainly show high results in the primary detection of brain tumor pathology. The use of a large number of pulse sequences allows differential diagnostics of detected pathologic changes, which is certainly a limitation of computed tomography performed in an emergency. The authors of the article analyzed a large amount of material on real clinical cases in patients with primary diagnosed brain tumor performed during the study in an urgent order. In this article it was shown that the findings of radiologists' conclusions about the nature of the mass according to urgently performed MRI do correlate with the subsequent final diagnosis of patients with high accuracy. However, with regard to the differential diagnosis of multiple or cystic masses, examination of other areas of the body, in particular CT of the thoracic cavity and CT/MRI of the abdomen, MRI of the pelvis, is required in order to search for the primary tumor. Thus, the judgment that chest CT in the case of a patient with a cystic brain mass was redundant is unfounded. Since at the primary stage it is necessary to exclude the presence of the underlying disease (primary tumor) as it directly affects the determination of treatment tactics in oncology. In general, when detecting brain masses suspicious for metastasis, including solitary ones, body screening for oncologic search is necessary in all cases. It is necessary to paraphrase these judgments in the relevant paragraphs. The second point that I think is important is that the factor of patients having HIV was not considered. This is relevant, especially in emergency medicine. There are many intracerebral manifestations of HIV, such as HIV encephalopathy, PML, HIV-associated encephalitis, Neurotoxoplasmosis, which can mimic metastatic brain lesions and in some cases are similar to lymphoma and primary gliomas. I wish the authors had considered this point too and added the information to the article. The article can certainly be accepted for publication after minor revisions.

Reviewer 2 Report

Comments and Suggestions for Authors

This manuscript evaluates the accuracy of radiologists in characterizing intra-axial brain tumors using emergency MRI reports in 150 patients. The findings show high accuracy for identifying gliomas, metastases, and lymphomas, with overall diagnostic accuracy deemed clinically acceptable. Despite some misclassifications, particularly in multifocal gliomas and metastases, the results suggest that emergency MRI can effectively guide initial diagnostic pathways. The study highlights the need for benchmarking human performance to evaluate the added value of AI models in this context.

There are some notable strengths of the manuscripts,

1.     Provides a practical assessment of radiologist performance in a real-world emergency setting, highlighting the challenges and potential solutions.

2.     Demonstrates high sensitivity and specificity for common intra-axial brain tumors, supporting the reliability of emergency MRI reports.

3.     Includes detailed statistical analysis and comparison across different radiologist experience levels and reporting times.

4.     Addresses a critical need for accurate and timely brain tumor characterization in emergency departments, with implications for patient management and treatment planning.

5.     Establishes a baseline for human diagnostic accuracy, which is essential for evaluating the potential benefits of AI applications in this field.

Few Questions / Comments on the manuscript are as follows,

1.     It would be good to know about the number of subjects that were rejected based on the exclusion criteria, specifically once with no final clinical diagnosis in patient charts and no specific suggestion of a tumor type on primary MRI report.

2.     The manuscript mentions using a Philips Ingenia 3-Tesla machine with various sequences (T2-weighted, FLAIR, DWI, SWI, T1 and T1 with gadolinium). Can the author provide more details on the specific parameters used for these sequences?

3.     Also, can the author mention regading the preprocessing steps, Was there any pre-processing steps applied to the MRI images, before reviewing the images?, Specifically were there any steps that were found helpful to the radiologist for viewing the MRI Images.

4.     Figure 1, It would be good if the authors could add figure description other than the existing figure name, mentioning more on this figure.

5.     Also, It would be good to understand more on the results, specifically how was the patient information, age, history of malignancy etc. compared to the clinical diagnosis of the results.

6.     The results section did provide metrics for sensitivity, specificity, PPV, NPV, and accuracy for gliomas, metastases, and lymphomas. It would be good if the author could provide additional metrics such as the F1-score or AUC-ROC to give a more comprehensive evaluation of the model's performance?

7.     Table 3, the result section, there was variability seen in the sensitivity, PPV, NPV for Glioma vs other pathologies, Metastasis vs other pathologies, and for lymphoma vs other pathologies. Can the author mention few points on this variability observed in the result section.

8.     The results were not provided with any standard deviation. The standard deviation value would help in understanding the results and its variance.

9.     The manuscript reports that radiologists with varying levels of experience (radiologists in training, board-certified radiologists, fellowship-trained neuroradiologists) performed the primary MRI reports. How did the differences in experience levels impact the accuracy of the tumor type suggestions? Were there any notable trends or variations?

10.  The manuscript found that the misinterpreted cases (Figure 2. a.) with multiple tumors were histopathologically confirmed gliomas, initially suspected as metastases. Can you provide more insights into the specific imaging features for misclassifications and how they might be addressed in future?

11.  Section for discussion mentions that the accuracy of emergency MRI reports is comparable to that of contemporary AI models. Can you provide more details on this comparison? Were specific AI models compared for this study, and how do the results align with those from AI-based approaches?

Reviewer 3 Report

Comments and Suggestions for Authors

This a well written study on the accuracy of radiological reports with regard to brain tumors in the emergency room setting. The authors report a satisfying accuracy. Introduction is rather lengthy.

My main point is the following one:

Honestly I do not understand the research question. Why should a radiologist’s report differ if it is given within or outside office hours? In the emergency room setting the only thing the clinician needs to know is whether there is something to do immediately. Anything else, i. e. the final radiology report that guides the further procedures, can be made during the regular office hours.

Minor points:

Line 116: These exclusion criteria lead to bias (overestimation of positive results) Why not inclusion of any mass effects?

Line 151: Why no nocturnal reports? Obviously there is no radiologists on call 24/7

Round 2

Reviewer 3 Report

Comments and Suggestions for Authors

I appreciate the response of the authors. If, as the authors state, "the emergence department has been changing towards an extension of the regular outpatient clinic in addition to taking care of real emergencies" the title of the paper should be changed, eg. "Accuracy of intra-axial brain tumor characterization outside regular office hours" or something like that. 

Author Response

Comment: I appreciate the response of the authors. If, as the authors state, "the emergence department has been changing towards an extension of the regular outpatient clinic in addition to taking care of real emergencies" the title of the paper should be changed, eg. "Accuracy of intra-axial brain tumor characterization outside regular office hours" or something like that. 

Response:

We thank the reviewer for the possibility of clarifying this subject.

We apologize for this misunderstanding caused by ambiguous wording in our previous response. While we speculated on the causes of increased complex emergency imaging, we did not intend to suggest that these cases would be considered elective. Our emergency MRI functions solely on an emergency basis with no elective imaging slots. All of the patients presented in this paper were imaged at the emergency department based on the request of the emergency physicians. Therefore, we would like to keep the title "...in the emergency MRI reports..." because it is fully consistent with the patients presented here.